# Nasal Cannula with Long and Narrow Tubing for Non-Invasive Respiratory Support in Preterm Neonates: A Systematic Review and Meta-Analysis

**DOI:** 10.3390/children9101461

**Published:** 2022-09-23

**Authors:** Pratima Anand, Monika Kaushal, Viraraghavan Vadakkencherry Ramaswamy, Abdul Kareem Pullattayil S., Abdul Razak, Daniele Trevisanuto

**Affiliations:** 1Division of Neonatology, Department of Paediatrics, Vardhman Mahavir Medical College and Safdarjung Hospital, New Delhi 110029, India; 2Department of Neonatology, Emirates Speciality Hospital, Dubai P.O. Box 505240, United Arab Emirates; 3Department of Neonatology, Ankura Hospital for Women and Children, Hyderabad 520072, India; 4Health Sciences Library, Queen’s University, Kingston, ON K7L 3N6, Canada; 5Division of Neonatology, Department of Paediatrics, Princess Nourah Bint Abdulrahman University, King Abdullah Bin Abdulaziz University Hospital, Riyadh 11564, Saudi Arabia; 6Department of Paediatrics, Monash University, Clayton, VIC 3800, Australia; 7Department of Woman and Child Health, University of Padua, University Hospital of Padua, 35128 Padua, Italy

**Keywords:** nasal cannula, neonate, noninvasive respiratory support, preterm infant, respiratory distress, systematic review

## Abstract

Background: Cannulas with long and narrow tubing (CLNT) are increasingly being used as an interface for noninvasive respiratory support (NRS) in preterm neonates; however, their efficacy compared to commonly used nasal interfaces such as short binasal prongs (SBP) and nasal masks (NM) has not been widely studied. Material and Methods: Medline, Embase, CENTRAL, Health Technology Assessment Database, and Web of Science were searched for randomized clinical trials (RCTs) and observational studies investigating the efficacy of CLNT compared to SBP or NM in preterm neonates requiring NRS for primary respiratory and post-extubation support. A random-effects meta-analysis was used for data synthesis. Results: Three RCTs and three observational studies were included. Clinical benefit or harm could not be ruled out for the outcome of need for invasive mechanical ventilation (IMV) for CLNT versus SBP or NM [relative risk (RR) 1.37, 95% confidence interval (CI) 0.61–3.04, certainty of evidence (CoE) low]. The results were also inconclusive for the outcome of treatment failure [RR 1.20, 95% CI 0.48–3.01, CoE very low]. Oropharyngeal pressure transmission was possibly lower with CLNT compared to other interfaces [MD −1.84 cm H20, 95% CI −3.12 to −0.56, CoE very low]. Clinical benefit or harm could not be excluded with CLNT compared to SBP or NM for the outcomes of duration of IMV, nasal trauma, receipt of surfactant, air leak, and NRS duration. Conclusion: Very low to low CoE and statistically nonsignificant results for the clinical outcomes precluded us from making any reasonable conclusions; however, the use of CLNT as an NRS interface, compared to SBP or NM, possibly transmits lower oropharyngeal pressures. We suggest adequately powered multicentric RCTs to evaluate the efficacy of CLNT when compared to other interfaces.

## 1. Introduction

Respiratory distress is a common occurrence in the neonatal period which warrants intensive care admission, and an appropriate strategy for respiratory support is critical to improve neonatal outcomes. There is widespread use of noninvasive respiratory support (NRS) including high-flow nasal cannula (HFNC), nasal continuous positive airway pressure (nCPAP), biphasic CPAP (BiPAP), nasal intermittent positive pressure ventilation (NIPPV), synchronized NIPPV, nasal high-frequency ventilation (nHFOV), and non-invasive neurally adjusted ventilatory (NIV NAVA), all of which may be used as a primary modality for respiratory support or for post-extubation in respiratory distress in neonates [1,2]. Amongst the various NRS modalities, NIPPV have been shown to be probably the most efficacious modality for preventing the need for invasive mechanical ventilation or treatment failure when used as a primary or post-extubation support [1,3].

Apart from the type of NRS, nasal interfaces for applying NRS also play an important role in preventing treatment failure. The resistance to flow and pressure transmission, and hence the clinical outcomes, may be influenced by the choice of interface [4]. It is also important to note that the force applied on the nasal skin does not compromise the skin integrity and result in nasal trauma [5,6,7,8,9,10,11]. In addition, the perceived comfort of the neonate and the ease of use for the health personnel are the other important factors that determine the choice of interface in many neonatal care units [3].

Short binasal prongs (SBP) [Hudson (Hudson RCI, Temecula, CA, USA), Fisher & Paykel (Fischer & Paykel Healthcare, Auckland, New Zealand) & Argyll (Cardinal Health, Dublin, OH, USA)], nasal masks (NM), and nasopharyngeal tubes are the common interfaces utilized for instituting NRS [12]. These interfaces have been studied for clinical efficacy over the last decade, with a recent meta-analysis indicating NM to be more effective when compared to SBP in preventing endotracheal intubation within 72 h of initiating nCPAP [13]. This meta-analysis also reported a reduced risk of nasal trauma with NM compared to SBP while delivering nCPAP support in preterm neonates. 

RAM Cannula (Neotech, Valencia, NM, USA), a type of cannula with long and narrow tubing (CLNT), has been approved by the Food and Drugs Administration (FDA) as a class-1 medical device for providing supplemental oxygen with 60% to 80% occlusion of the nares [14]. It is used off-label in many NICUs as an interface for nCPAP and NIPPV, as it is perceived to be associated with less discomfort and nasal trauma [12,14,15]. Simulation-based lung studies on pressure transmission and tidal volume generation using CLNT have reported equivocal results [16,17,18,19,20]. Some of the studies have shown that resistance to air flow is higher with CLNT than with SBP [18]. There is a lot of ambiguity in the published literature with respect to pressure transmission with CLNT [19,21]. Similarly, results from clinical studies on CLNT in preterm neonates show conflicting data on various outcomes such as treatment failure and need for invasive mechanical ventilation (IMV) [21,22,23,24,25,26]. It is therefore essential that the literature related to CLNT is systematically evaluated to derive meaningful conclusions about its use in clinical practice. This meta-analysis is hence aimed to study the effect of any NRS given with CLNT, otherwise referred to as RAM cannula, compared with SBP, NM, or both on the various outcomes in preterm neonates requiring NRS for initial treatment or for post-extubation.

## 2. Materials and Methods

The protocol was registered with PROSPERO (CRD42021226178 available at https://www.crd.york.ac.uk/prospero/ accessed on 1 September 2022), and the reporting of this systematic review is in accordance with the preferred reporting items for systematic reviews and meta-analyses (PRISMA) framework [27].

### 2.1. Literature Search

The following electronic databases were searched from the inception until 7 July 2022: Medline, Embase, The Cochrane Library (Cochrane Database of Systematic Reviews, Cochrane Central Register of Controlled Trials (CENTRAL), the Cochrane Methodology Register), the Health Technology Assessment Database, and the Web of Science. There were no language restrictions. Two authors (M.K. and P.A.) performed the literature search independently. References in the included studies and trial registries such as World Health Organization (WHO) and ClinicalTrials.gov were also searched. Conference abstracts of pediatric academic societies from the last three years were searched and were included only if sufficient data was available for risk-of-bias assessment. Lastly, citations of the included studies were also searched for possible inclusion. The literature search strategy for all the databases is provided in Appendix A.

### 2.2. Inclusion Criteria

Randomized controlled trials (RCTs), quasi-RCTs, crossover trials, and observational studies were eligible for inclusion. Descriptive studies with no control arm, narrative reviews, case reports, case series, commentaries, and letters to editors were excluded.

Population: Preterm neonates (<37 weeks’ gestation) requiring NRS modalities of NIPPV or NCPAP as primary or post-extubation support were included.
Intervention: CLNTComparators: SBP or NM

### 2.3. Outcomes

#### 2.3.1. Primary Outcome

Treatment failure was defined as the need for escalation to a higher mode of NRS or the need for invasive mechanical ventilation (IMV) at any time within the first 7 days of initiating the support. The indication and criteria for escalation of respiratory support was as defined by the study authors. As treatment failure and the need for IMV are not mutually exclusive, we considered treatment failure and the need for IMV separately as primary outcomes.

#### 2.3.2. Secondary Outcomes

Clinical outcomes: receipt of surfactant therapy, air leak (as defined by authors), nasal trauma occurring at any time until the discontinuation of respiratory support (any grade and severity, as defined by the authors), and duration of IMV and NRS (days).Surrogate outcomes: pressure transmission at the level of the pharynx or esophagus and the work of breathing (using an objective and validated scoring technique).

### 2.4. Data Extraction, Data Synthesis, and Quality of Evidence

Data extraction was performed by two authors independently using a prespecified structured proforma. R-software (version 3.6.2) (R Foundation for Statistical Computing, Vienna, Austria) was used for the meta-analysis [28]. The Mantel–Haenszel method and the inverse variance method for dichotomous outcomes and continuous outcomes were utilized, respectively. Between the studies, heterogeneity was evaluated using Cochran Q, I2, and τ2 values. The random-effects model was preferred over the fixed-effects model; this was owing to the fact that clinical heterogeneity was anticipated between the studies, as some of the studies had enrolled neonates in whom NRS was used as a primary modality, while others used a mixture of primary as well as post-extubation modality. Also, the type of respiratory support varied, with some utilizing CPAP and others NIPPV.

The overall effect estimate for each of the outcomes was expressed as risk ratios (RR) and risk difference (RD) for dichotomous outcomes and mean differences (MD) for continuous outcomes, with their 95% confidence interval (CI) depicted using forest plots. The within-group standard error of the mean (SEM) reported in a trial was converted to the corresponding standard deviation (SD) [29]. Publication bias was planned to be assessed by using funnel plots if we included 10 or more clinical trials in the meta-analysis.

### 2.5. Risk-of-Bias (RoB) Assessment

Cochrane RoB tool version 2.0 was utilized for RCTs [30] and Risk of Bias in Non-randomized Studies of intervention (ROBINS-I) for non-RCTs [31]. Two authors (V.R. and A.R.) assessed the RoB independently, and conflicts were resolved by consensus and discussion.

### 2.6. Certainty of Evidence (CoE) Assessment

Grading of recommendations, assessment, development, and evaluations (GRADE) was used for assessing the CoE, which was rated as high, moderate, low, or very low [32,33,34]. The results of the meta-analysis were reported as per a modified GRADE working group recommendation [35] (Appendix A).

### 2.7. Subgroup Analysis

The studies were pooled as subgroups depending on the type of NRS and the indication for NRS for investigating the heterogeneity:i.type of NRS: NCPAP vs. NIPPVii.indication for NRS: primary, post-extubation, or both

We intended to pool the results based on the gestational age of the neonates: <28 weeks and ≥28 weeks if two or more studies provided separate data for the relevant subgroup.

## 3. Results

Of the 105 studies identified during the systematic search of the literature, a total of six studies were included (RCTs: 3 studies (Maram 2021, Hochwald 2021, and Gokce 2021) (*n* = 521); [24,25,26] observational studies: 3 studies (Singh 2018, Sharma 2020, Drescher 2018) (*n* = 138)) [21,23]. The PRISMA flow is depicted in Figure 1. The three RCTs had enrolled neonates born at a gestational age of 28–34 weeks [24,25,26]. Two of the RCTs (Gokce 2021 and Hochwald 2021) [24,25] used NIPPV as the NRS modality, and one (Maram 2021) used CPAP [26]. Two of the RCTs (Maram 2021 and Gokce 2021) [24,26] included enrolled neonates who were instituted NRS as primary respiratory support, and one (Hochwald 2021) used it both as primary and as post-extubation [25]. The characteristics of the included studies are given in Table 1. The reasons for the studies that were excluded are given in Appendix A [4,17,19,21,36,37,38,39]. The comparator in four of the six included studies was SBP [18,23,24,25,26]. In the remainder of the studies, multiple interfaces were used, including SBP [22,25].

### 3.1. RoB of the Included Studies

Of the three RCTs included, one provoked concern due to issues in the domain randomization process (Gokce 2019), and the other two RCTs (Hochwald 2021, Maram 2019) had a low risk of overall bias [24,25,26]. For these RCTs, the randomization process was Web-based, deviations from intended interventions were minimal, the measured outcomes were objective, there was no missing data, and all the selected outcomes were reported.

Among the non-randomized studies, two (Sharma 2020 and Drescher 2018) had a serious risk of overall bias predominantly contributed by confounding, and one had moderate risk of bias in the domain of measurement of outcomes (Singh 2018) [18,23]. The risk of bias in the included studies is given in Appendix A (randomized trials) and Appendix A (non-randomized trials).

### 3.2. Outcomes from Randomised Trials

#### 3.2.1. Primary Outcomes

##### Treatment Failure

Clinical benefit or harm could not be ruled out for the outcome of treatment failure between the two groups [RR 1.20, 95% CI 0.48–3.01; 3 studies, 521 participants]. The CoE was downgraded to very low certainty due to limitations in the study design, indirectness, inconsistency, and imprecision (Figure 2, Table 2). The test for subgroup differences was not significant.

##### Need for IMV

The meta-analysis showed that clinical benefit or harm could not be ruled out with CLNT compared with NRS provided with SBP [RR 1.37, 95% CI 0.61–3.04; risk difference 23 more per 1000, 95% CI 20–125 more; 3 studies, 521 participants]. The CoE was rated down by two levels to low due to serious limitations of inconsistency and imprecision (Figure 3, Table 2). The test for subgroup differences was not significant.

### 3.3. Secondary Outcomes

#### 3.3.1. Nasal Trauma

Clinical benefit or harm could not be ruled out for the outcome of nasal trauma between the two groups [RR 0.49, 95% CI 0.21–1.11; 3 studies, 521 participants; CoE: low]. The test for subgroup differences for NRS modes, NCPAP and NIPPV, was significant. (Figure 3, Table 2).

#### 3.3.2. Surfactant Treatment

Clinical benefit or harm could not be ruled out for the outcome of surfactant therapy between the two groups [RR 1.44, 95% CI 0.68–3.04; 2 studies, 292 participants; CoE: very low] (Figure 2, Table 2). The test for subgroup differences was significant.

#### 3.3.3. Duration of IMV

The meta-analysis of 2 studies involving 292 participants could not provide any meaningful interpretation with the use of CLNT when compared to other interfaces [mean difference (MD) of 5.07 days and 95% CI of −1.04 to 11.19 days]. The CoE was rated down to very low certainty due to serious limitations in the study design, serious risk of bias, high heterogeneity, and imprecision (Figure 3, Table 2). The test for subgroup differences was significantly different.

#### 3.3.4. Duration of NRS

Clinical benefit or harm could not be ruled out for the outcome of duration of NRS between the two groups [MD 2.85 days, 95% CI −0.95 to 6.64; 3 studies, 521 participants], and CoE was very low (Figure 4, Table 2).

#### 3.3.5. Air Leak

Clinical benefit or harm could not be ruled out for the outcome of air leak between the two groups [RR 1.20, 95% CI 0.36–4.00; 3 studies, 521 participants], and the CoE was low (Figure 3, Table 2).

#### 3.3.6. Work of Breathing (Using Objective and Validated Scoring Technique) and Pressure Transmission (at the Level of Pharynx or Oesophagus)

None of the included RCTs evaluated this outcome.

### 3.4. Outcomes from Non-Randomised Trials

Three observational trials (Singh 2018, Sharma 2020, and Drescher 2018) which included 138 participants satisfied the eligibility criteria for this systematic review [18,22,23].

#### 3.4.1. Primary Outcomes

##### Primary Outcomes

No meta-analysis was performed, as only one study reported treatment failure between the groups. The treatment failure reported by Drescher et al. was not statistically different between the two groups [RR 2.81, 95% CI 0.84 to 9.36]

##### Secondary Outcomes

*Oropharyngeal pressure transmission*: The oropharyngeal pressure transmission was possibly lower with CLNT when compared to other nasal interfaces [MD −1.84 cm H20, 95% CI −3.12 to −0.56; 2 studies, 106 participants; CoE: very low] (Figure 4, Table 2).

No meta-analysis was performed for other secondary outcomes, as only a single observational study reported on some of these outcomes. This study (Drescher 2018) provided data for nasal trauma, IMV duration, and NRS duration. The IMV duration was significantly higher with CLNT with a mean difference of 7.58 days and 95% CI of 0.32 to 14.84. The same study reported nasal trauma as one of the secondary outcomes, which was significantly lower with CLNT [RR of 0.16, 95%CI 0.05 to 0.49, CoE: very low]. Duration of NRS was significantly lower with CLNT with a mean difference of 8.70 days and 95% CI of −15.88 to −1.52.

## 4. Discussion

In this systematic review and meta-analysis, we evaluated the efficacy of CLNT in comparison to other routinely utilized nasal interfaces for NRS, such as SBP and NM. To the best of our knowledge, this is the only systematic review and meta-analysis highlighting the evidence from six available clinical studies, three randomized (*n* = 521) and three non-randomized (*n* = 138), evaluating the efficacy of CLNT as an interface for NRS in preterm neonates for primary respiratory support or for post-extubation. Though CLNT possibly results in lower oropharyngeal pressure transmission when compared to SBP/NM (meta-analysis of non-RCTs), clinical benefit or harm could not be ruled out for most of the outcomes evaluated, highlighting the need for future trials on this question. Our finding of lower oropharyngeal pressure transmission with CLNT is consistent with the earlier in vivo and bench studies on CLNT showing lower pressure delivery with CLNT [4,15,16,17,19,20,21,36]. The RAM cannula is quite similar to the traditional nasal prongs used to provide supplemental oxygen. Its long and narrow tubing is supposed to be more effective than traditional cannula in transmitting pressures to the nasal end, which in turn is relatively soft compared to SBPs [24,25,26]. Due to the very low CoE, the findings of our meta-analysis need to be validated by future studies.

Of the primary outcomes evaluated, treatment failure and need for IMV were assessed separately, as these outcomes are not mutually exclusive and neonates who meet the criteria for treatment failure may be supported with a higher mode of NRS rather than being initiated on IMV. We found no significant differences for both outcomes; however, the confidence limits included a significant benefit and harm, thereby obviating the ability to draw any meaningful conclusions. The softer nasal cannula which comes in contact with the nares is perceived to be associated with lesser nasal trauma compared to SBPs or NM [24,25,26]. We evaluated nasal trauma as one of the secondary outcomes, and it is usually one of the determining factors for the preferential choice of a specific NRS interface in most of the neonatal units. Although our meta-analysis indicated no difference in nasal injury with the use of CLNT cannula, clinical benefit or harm could not be ruled out due to an imprecise effect estimate and low certainty of the evidence. The between-studies heterogeneity in this meta-analysis for the outcomes could possibly be explained by the differences in the patient population, co-interventions, baseline event rates, and the variable assessment methods and criteria used to evaluate the outcomes [24,25,26]. Other factors to be considered include the nurse-to-patient ratio, the skill and training of the healthcare personnel, and intensity of monitoring, all of which are known to influence the important outcomes.

Our study has several strengths. To date, to the best of our knowledge, this is the only systematic review and meta-analysis summarizing the current evidence on a pertinent question of clinical importance related to the use of an increasingly adopted NRS interface, CLNT, in preterm neonates for primary and post-extubation support. The study followed the standard methodology per the PRISMA framework, the Cochrane group guidance for systematic reviews including a comprehensive search strategy, prospective protocol registration in PROSPERO, and explicitly defined the clinical question and synthesized the evidence using appropriate methods. Lastly, our review also examines the CoE using the GRADE working group guidelines, an essential aspect to appraise the quality and strength of evidence and thereby making an informed clinical decision.

However, we acknowledge some of the limitations of this meta-analysis. Firstly, the number of studies contributing to the systematic review and meta-analysis is limited. Secondly, despite synthesizing the evidence in a meta-analysis, the confidence in the evidence for many outcomes studied is limited due to very low to low evidence certainty. We identified a few ongoing trials (CTRI NCT03121781, NCT0216825, and CTRI/2020/03/024097) that may likely address these limitations [40,41,42]. We also recognize that there was a lack of homogeneity among the studies about the type of NRS used as well as its indication of use (primary support versus post-extubation), comparator interfaces, varying definitions, and assessment of primary and secondary outcomes, as evident by statistical heterogeneity for many outcomes. We explored the heterogeneity by pooling the studies as subgroups based on the type and indication of NRS and found the test for subgroup differences was statistically significant for some outcomes suggesting one of possible reasons for the observed heterogeneity. We also attempted to study the subgroup data based on the gestational age of the included neonates: (<28 weeks versus >28 weeks). However, limited reporting in the included RCTs or observational studies precluded such analyses.

## 5. Conclusions

The results of this meta-analysis indicate that compared to SBP or NM, the use of CLNT as an interface for NRS either as a primary support modality or for post-extubation in preterm neonates possibly results in reduced oropharyngeal pressure transmission; however, the meta-analysis could not assess with certainty the effect on important clinical outcomes. Henceforth, we suggest this as priority research and until more high-quality evidence is available, clinicians should consider proven nasal interfaces, such as NM or SBP, that are likely to provide effective pressure transmission, adequate ventilation, and oxygenation, thereby possibly avoiding IMV in preterm neonates.

## Figures and Tables

**Figure 1 children-09-01461-f001:**
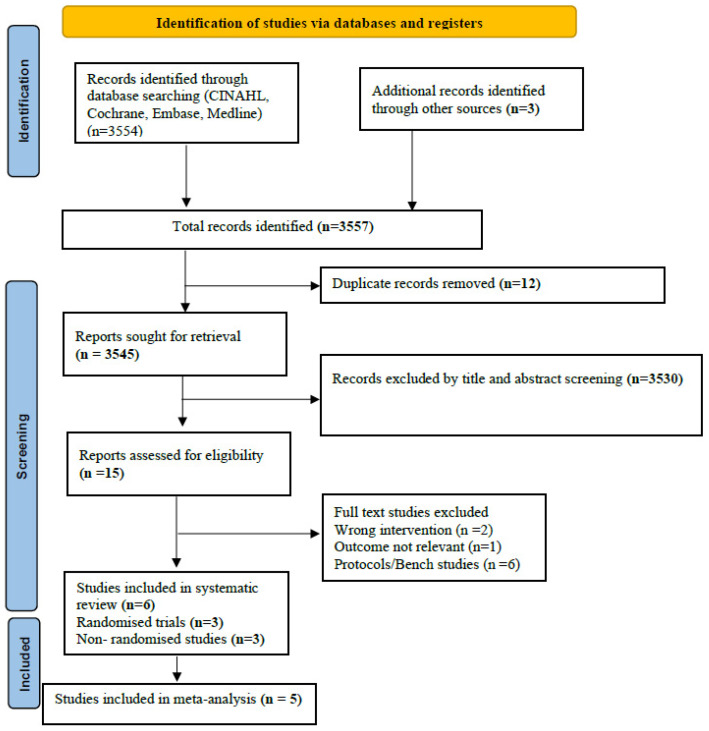
PRISMA flow.

**Figure 2 children-09-01461-f002:**
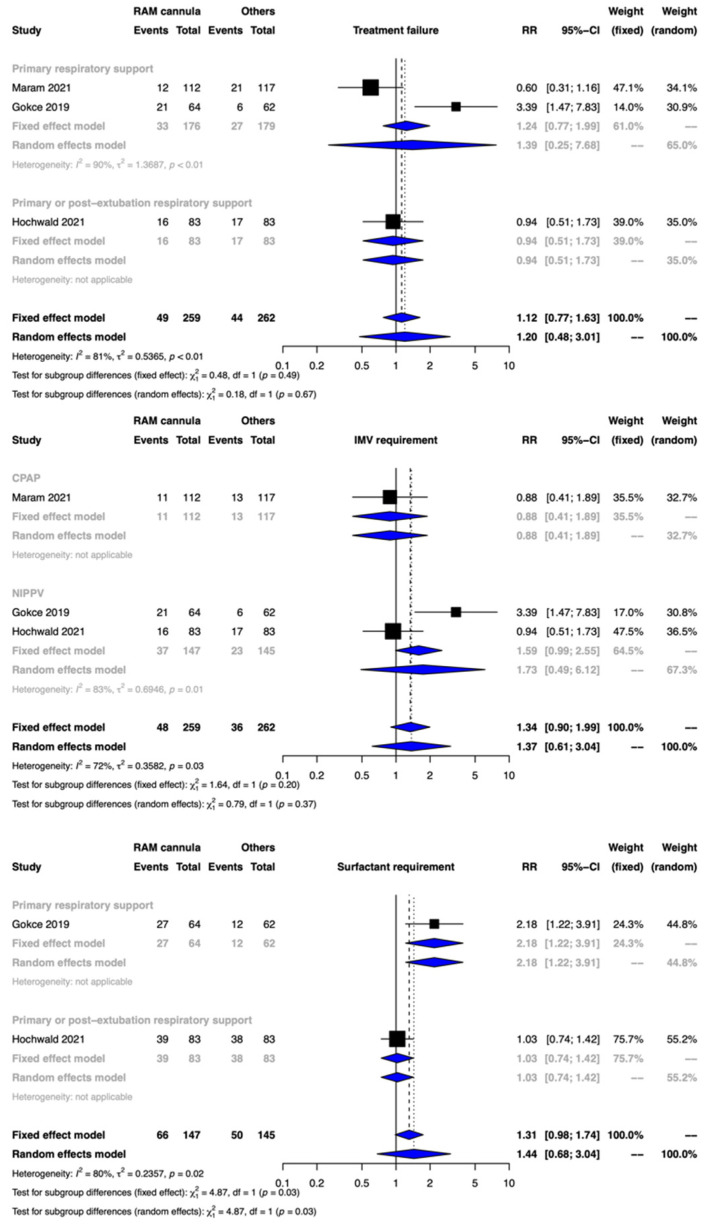
Forest plots depicting the effect estimates for the outcomes: treatment failure, invasive mechanical (IMV) requirement, and surfactant requirement.

**Figure 3 children-09-01461-f003:**
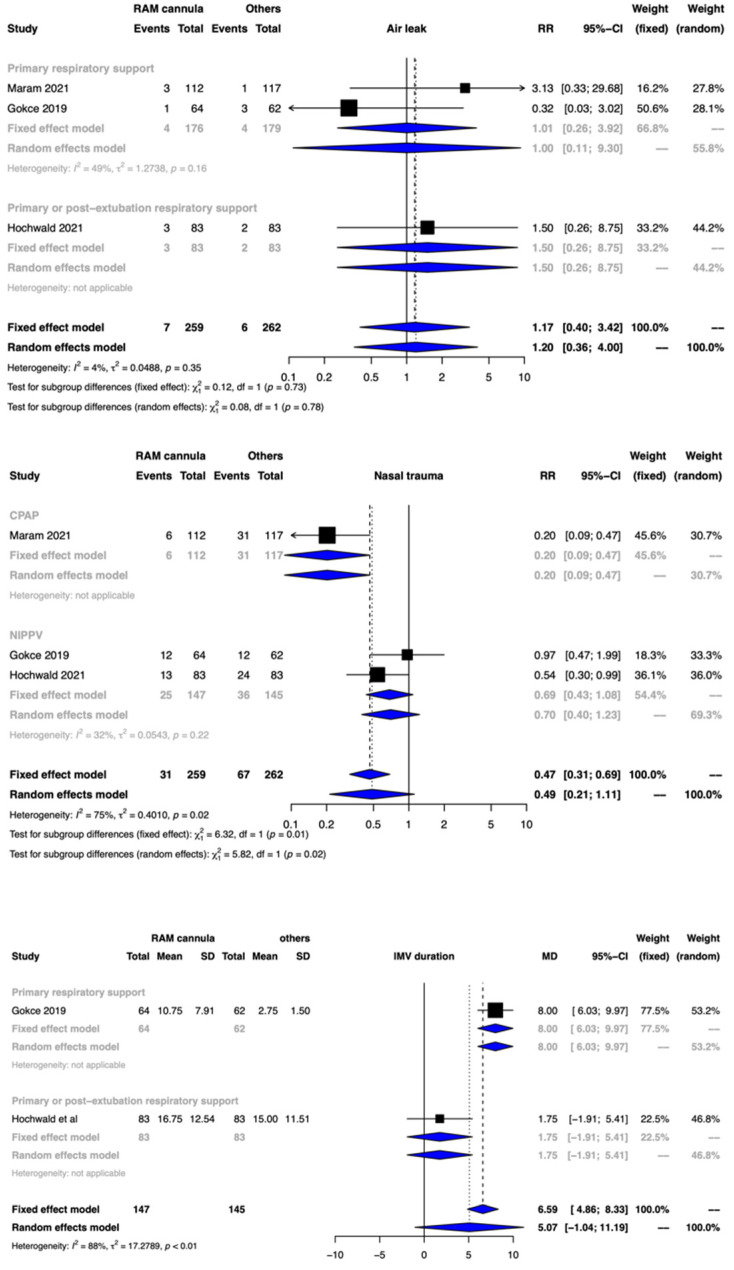
Forest plots depicting the effect estimates for the outcomes: air leak, nasal trauma, and invasive mechanical (IMV) duration.

**Figure 4 children-09-01461-f004:**
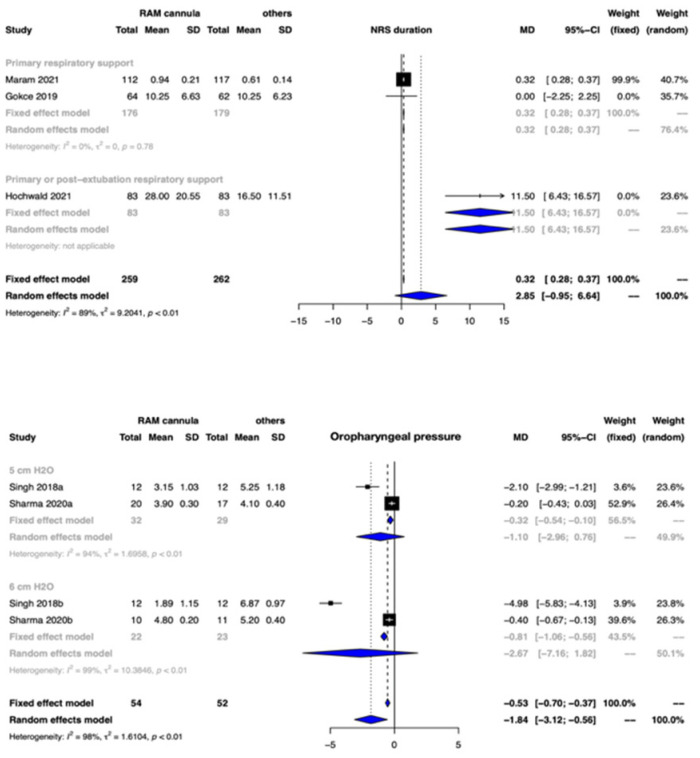
Forest plots depicting the effect estimates for the outcomes: noninvasive respiratory support (NRS) duration and oropharyngeal pressure transmission.

**Table 1 children-09-01461-t001:** Characteristics of the included studies.

Study ID, Year, Country	Intervention	Comparator	GA (Weeks) Mean ± SD or Median (IQR)	BW (g) Mean ± SD or Median (IQR)	Type of NRS Used	Study Outcomes	Other Comments
Maram et al. 2021 India [26]	CPAPwith RAM cannulae with cannulaide	CPAP with short binasal prongs	I: 31.4 ± 1.7 C:31.4 ± 1.6	I: 1491 ± 321 C: 1531 ± 394	CPAP as primary respiratory support	Primary outcome: Incidence and severity of nasal injury Secondary outcomes: Nasal injury score at discharge Need for IMV Duration of nasal CPAP Need for change in interface Mortality CPAP failure Culture positive sepsis PDA NEC IVH grade 3 or more Cystic PVL ROP needing laser Supplemental oxygen at 28 days Air leak Transfer to other hospital Discharge from hospital Length of hospital stay Weight at discharge Length at discharge Head circumference at discharge	Inclusion criteria: Neonates between 28 to 34 weeks of gestation, stratified based on GA 28 to 30 weeks and 31 to 34 weeks Exclusion criteria: Neonates who required IMV at admission to NICU; those with poor respiratory efforts or apnea, worsening shock, suspected or proven persistent pulmonary hypertension of newborn, severe metabolic acidosis (pH < 7.20 and base deficit > 10), severe respiratory acidosis (pH < 7.20 and PaCO2 > 60 mm Hg), and massive pulmonary haemorrhage. Neonates with major congenital malformations such as congenital diaphragmatic hernia, tracheo-oesophageal fistula, Pierre Robin sequence, and choanal atresia were excluded.
Gokce et al. 2019 Turkey [24]	NIPPV(non-synchronised) using RAM cannula	NIPPV using short binasal prongs	All: 29.6 ± 2.0 I: 29.5 ± 2.2 C: 29.8 ± 1.8	All: 1254 ± 348 I: 1255 ± 348 C: 1253 ± 350	Non synchronised NIPPV as primary support	Primary outcome: Need for IMV (failure of NIPPV) within the first 72 h of life Need for surfactant Secondary outcomes: Duration of NIPPV Duration of IMV Duration of supplemental oxygen Incidence of nasal injury: nasal septal injury stage 1 or 2 NEC IVH > grade 2 Pneumothorax Pulmonary interstitial emphysema Pulmonary haemorrhage BPD Duration of hospital stay Death	Inclusion criteria: Neonates between 26 to 33^+6^ weeks gestation stratified from 26 weeks to 29^+6^ weeks and 30 to 33 ^+6^ weeks Exclusion criteria: Infants who required intubation in the delivery room, those with a major congenital anomaly, or those were transferred to hospital after birth in another center.
Hochwald et al. 2021 Israel [25]	Synchronised NIPPV using RAM cannula	NIPPV with short binasal prongs	I: 29.3 ± 2.2 C: 29.2 ±2.5	I: 1237 ± 414 C: 1254 ±448	Synchronised NIPPV (Primary respiratory support and post extubation)	Primary outcome: Treatment failure within 72 h after initiation of NIPPV, i.e., need for IMV	Inclusion criteria: Neonates between 24 to 33^+6^ weeks gestation Exclusion criteria: Infants with significant morbidity apart from RDS, including cardiac disease (not including patent ductus arteriosus), congenital malformation, or if they had cardiovascular or respiratory instability because of sepsis, anaemia, or severe IVH.
Singh et al. 2021 U.S.A [23]	CPAP using RAM cannula	CPAP using short binasal prongs	28.1 ± 2.1	1225 ± 405	CPAP	Intraoral (pharyngeal) pressures	Inclusion criteria: Any preterm infant with respiratory distress requiring CPAP but not IMV or NIPPV Exclusion criteria: Critically ill or had major congenital anomalies, neuromuscular disorders, or upper airway anomalies.
Sharma et al. 2020 India [18]	CPAP using RAM cannula	CPAP using nasal mask or short binasal prongs	I: 32 (29 to 33) C: 32(29 to 33)	I: 1331 ± 228 C: 1382 ± 209	CPAP	Mean pharyngeal pressure	Inclusion criteria: Preterm neonates with gestation 28 to 34 weeks and BW more than or equal to 1000 g and requiring nasal CPAP for respiratory distress. Exclusion criteria: infants who were critically ill and those who had major congenital anomalies or upper airway malformations.
Drescher et al. 2018 USA [22]	NRS which included CPAP, NIPPV and HHHFNC using RAM cannula (with barrier nursing).	NRS using all other interfaces before the implementation of RAM system as interface in 2014. (Historical control).	I: 26.9 ± 1.3 C: 26.4 ± 1.6	I: 879 ±192.9 C: 866.6 ± 185.4	NIPPV (Primary respiratory support and post extubation)	Total IMV courses Total IMV duration Initial IMV duration Successful extubation Time until initial re-intubation Total re-intubations NRS failure < 7 days Total duration of CPAP Total duration of NIPPV Total duration of HFNC Total duration of nasal cannula use Total duration of NRS Total duration of respiratory support BPD Skin and/or mucosal breakdown	Inclusion criteria: Neonates < 1500 g and/or less than 29 weeks, gestation Exclusion criteria: any infant who died before the initiation of NRS or who was transferred before 36 weeks’ postmenstrual age.

GA: gestational age, BW: birth weight, SD: standard deviation, IQR: interquartile range, NRS: noninvasive respiratory support, CPAP: continuous positive airway pressure, NEC: necrotising enterocolitis, PDA: patent ductus arteriosus, IVH: intraventricular hemorrhage, PVL: periventricular leukomalacia, ROP: retinopathy of prematurity, BPD: bronchopulmonary dysplasia, IMV: invasive mechanical ventilation, NIPPV: noninvasive positive pressure ventilation, HFNC: high flow nasal cannula, I: intervention, C: comparator.

**Table 2 children-09-01461-t002:** Certainty of evidence assessment for RAM cannula compared to other interfaces (binasal prongs and/or nasal mask) in preterm neonates requiring noninvasive respiratory support.

Certainty Assessment							Summary of Findings				
Participants (Studies) Follow-Up	Risk of Bias	Inconsistency	Indirectness	Imprecision	Publication Bias	Overall Certainty of Evidence	Study Event Rates (%)		Relative Effect (95% CI)	Anticipated Absolute Effects	
							With Other Interfaces (Binasal Prongs and/or Nasal Mask)	With RAM Cannula		Risk with other Interfaces (Binasal Prongs and/or Nasal Mask)	Risk Difference with RAM Cannula
**Treatment failure**											
521 (3 RCTs)	not serious	very serious ^a^	serious ^b^	serious ^c^	none	⨁◯◯◯ Very low	44/262 (16.8%)	49/259 (18.9%)	**RR 1.20** (0.48 to 3.01)	168 per 1000	**34 more per 1000** (from 87 fewer to 338 more)
**Invasive mechanical ventilation requirement**											
521 (3 RCTs)	not serious	serious ^d^	not serious ^e^	serious ^f^	none	⨁⨁◯◯ Low	16/262 (6.1%)	36/259 (13.9%)	**RR 1.37** (0.67 to 3.04)	61 per 1000	**23 more per 1000** (from 20 fewer to 125 more)
**Surfactant requirement**											
292 (2 RCTs)	not serious	very serious ^a^	serious ^b^	serious ^c^	none	⨁◯◯◯ Very low	66/145 (45.5%)	66/147 (44.9%)	**RR 1.44** (0.68 to 3.04)	455 per 1000	**200 more per 1000** (from 146 fewer to 929 more)
**Air leak**											
521 (3 RCTs)	not serious	not serious	serious ^b^	serious ^c^	none	⨁⨁◯◯ Low	6/262 (2.3%)	7/259 (2.7%)	**RR 1.20** (0.36 to 4.00)	23 per 1000	**5 more per 1000** (from 15 fewer to 69 more)
**Duration of invasive mechanical ventilation**											
292 (2 RCTs)	serious ^g^	very serious ^a^	not serious ^e^	serious ^h^	none	⨁◯◯◯ Very low	145	147	-		MD **5.07 days higher** (1.04 lower to 11.19 higher)
**Duration of non-invasive respiratory support**											
521 (3 RCTs)	not serious ^i^	very serious ^a^	not serious ^e^	serious ^h^	none	⨁◯◯◯ Very low	262	259	-		MD **2.85 days higher** (0.95 lower to 6.64 higher)
**Oropharyngeal pressure**											
106 (2 observational studies)	serious ^j^	very serious ^a^	not serious	serious ^f^	none	⨁◯◯◯ Very low	52	54	-		MD **1.84 cm H_2_O lower** (3.12 lower to 0.56 lower)
**Nasal Trauma**											
521 (3 RCTs)	not serious	serious ^a^	not serious	serious ^c^	none	⨁⨁◯◯ Low	67/262 (25.6%)	31/259 (12.0%)	**RR 0.49** (0.21 to 1.11)	256 per 1000	**130 fewer per 1000** (from 202 fewer to 28 more)

**Explanations:** ^a^ I2 > 75%. ^b^ Indirectness related to patient population as Hoschwald et al. study had enrolled neonates who required noninvasive respiratory support as a primary as well as a post-extubation mode. ^c^ 95% CI crossing line of no effect and Optimal Information Criterion (OIS) not satisfied due to low sample size and low event rates. ^d^ I2 > 50%. ^e^ Although there was indirectness related to patient population in the study of Hoschwald et al. as it had enrolled neonates who required noninvasive respiratory support as a primary as well as a post-extubation mode, the weightage for this study was minimal. Hence, the certainty of evidence was not downrated for indirectness. ^f^ OIS not satisfied. ^g^ Of the two studies, the study by Gocke et al., which had the highest weightage, had some concerns. ^h^ 95% CI crossing line of no effect. ^i^ Of the three RCTs, though the study by Gocke et al. had some concerns in risk of bias overall, the weightage of this study was 35.7%. Hence, the certainty of evidence was not downrated for risk of bias. ^j^ Of the two observational studies, that of Sharma et al. had a serious risk of overall bias. Certainty of evidence levels: +very low, ++ low, +++ moderate, ++++ high.

## Data Availability

Original data are available upon reasonable request to the corresponding author.

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
