# Peer review of "Nasal Cannula with Long and Narrow Tubing for Non-Invasive Respiratory Support in Preterm Neonates: A Systematic Review and Meta-Analysis"

_children, 2022, doi:10.3390/children9101461_

Round 1

Reviewer 1 Report

I reviewed the manuscript by Pratima Anand et al with great interest and would like to thank for the opportunity to review this manuscript.

This is a well conducted systematic review and meta-analysis, in which the efficacy of nasal cannula with long and narrow tubing (RAM cannula) in comparison to other routinely utilised nasal interfaces for non-invasive respiratory support, such as short binasal prongs and nasal masks. This systematic review and meta-analysis highlight the evidence from six available clinical studies, three randomised controlled trials (n=521) and three non-randomised studies (n=138), evaluating the efficacy of RAM cannula as an interface for non-invasive respiratory support in preterm neonates for primary respiratory support or for post-extubation. Though RAM cannula possibly results in lower oropharyngeal pressure transmission when compared to short binasal prongs and nasal masks (meta-analysis of non-RCTs), clinical benefit or harm could not be ruled out for most of the outcomes evaluated, highlighting need for future trials on this question.

Minor revisions need to be done before the manuscript could possibly be published.

Abstract:

Line 28: Change to:

…benefit or harm could not be ruled out for the outcome of… …need for invasive mechanical ventilation… or invasive mechanical ventilation requirement

Introduction:

Lines 61-62: Include the certain companies and headquarters!

Lines 73-74: Rephrase! …,and also because of its ease of use!?!?

Table 1: This table needs significant improvement and proof reading!

a), b), c) … is not necessary, and the Hochwald study didn`t receive a letter!

There is no list for the explanation of the abbreviations for this table: I?, C?, BW?, GA,…

For some interfaces companies and location of headquarters are mentioned in brackets, but not everywhere. e.g.  Leoni and SLE 5000 is not further disclosed. Neotech is sometimes mentioned for RAM, but sometimes not.

The presentation format of gestational age differs between studies, this would benefit from unification

The presentation format of BW is different between studies. Delete decimal numbers.

Maram et al.: Ram should be RAM in consistency to elsewhere

Gokce et al: Delete “the second outcomes were”; Delete numbers from 1 to 12.

Hochwald et al: What is NIPPC?

Neetu Singh – elsewhere only the last name is mentioned. This study was performed in India, not in Lebanon or New Hampshire

Sharma et al.: Draeger is spelled differently. Include a space after “C: 32” and “1382 ±

Drescher et al: Delete Medstar since the institutions were not mentioned in the other studies. Include DC, USA.  Delete 4x “median (range)”, “or” instead of “0r” 0 is a zero not an O.

Figure 1,2,3 low resolution, hard to read

There are two headings of Table 2, one in line 182 “Table 2. Certainty of evidence for all outcomes”, but a second within the table “Certainty of evidence assessment for RAM cannula”

Table 2 has a very bad resolution, reviewing this table was not possible due to the low quality.

Line 232-233 Please rephrase “as no two or more studies” makes no sense?!

The study is well conducted and the conclusions drawn by the authors are plausible.

Author Response

Response: We thank the Reviewer for the comments.

Abstract:

Line 28: Change to:

…benefit or harm could not be ruled out for the outcome of… …need for invasive mechanical ventilation… or invasive mechanical ventilation requirement

 Response: We thank the Reviewer for the suggestion and we have changed the sentence accordingly in the revised manuscript.

Introduction:

Lines 61-62: Include the certain companies and headquarters!

Response: We thank the Reviewer for pointing this out. We have included the details in the revised manuscript. (Page 2, lines 62-64)

Lines 73-74: Rephrase! …,and also because of its ease of use!?!?

Response: We have rephrased the sentence in the revised manuscript. (Page 2, lines 73-74)

 Table 1: This table needs significant improvement and proof reading!

a), b), c) … is not necessary, and the Hochwald study didn`t receive a letter!

There is no list for the explanation of the abbreviations for this table: I?, C?, BW?, GA,…

Response: We thank the Reviewer for indicating these. We have formatted Table 1. “a,b,c” have been removed. Abbreviations are explained in the Table footnotes.

For some interfaces companies and location of headquarters are mentioned in brackets, but not everywhere. e.g.  Leoni and SLE 5000 is not further disclosed. Neotech is sometimes mentioned for RAM, but sometimes not.

The presentation format of gestational age differs between studies, this would benefit from unification

The presentation format of BW is different between studies. Delete decimal numbers.

Maram et al.: Ram should be RAM in consistency to elsewhere

Gokce et al: Delete “the second outcomes were”; Delete numbers from 1 to 12.

Hochwald et al: What is NIPPC?

Response: We thank the Reviewer for pointing these out. The company names have been removed for the sake of uniformity; format of gestational age, birth weight, RAM cannula have been presented in similar formats across studies in the revised manuscript; NIPPC has been changed to NIPPV.

Neetu Singh – elsewhere only the last name is mentioned. This study was performed in India, not in Lebanon or New Hampshire

Response: We thank the Reviewer for the comment. We changed to “Singh”. The study was conducted in Lebanon, New Hampshire, U.S.A according to the information given in the paper.

Sharma et al.: Draeger is spelled differently. Include a space after “C: 32” and “1382 ±”

Drescher et al: Delete Medstar since the institutions were not mentioned in the other studies. Include DC, USA.  Delete 4x “median (range)”, “or” instead of “0r” 0 is a zero not an O.

 Response: We have made the suggested changes in the revised manuscript.

Figure 1,2,3 low resolution, hard to read

 Response: We have uploaded newer versions of Figures 1, 2, 3, 4 in the revised manuscript.

There are two headings of Table 2, one in line 182 “Table 2. Certainty of evidence for all outcomes”, but a second within the table “Certainty of evidence assessment for RAM cannula”

Table 2 has a very bad resolution, reviewing this table was not possible due to the low quality.

Response: Table heading which was repeating has been corrected and a higher resolution version of the Table 2 has been uploaded.

Line 232-233 Please rephrase “as no two or more studies” makes no sense?!

Response: We thank the reviewer for the comments. We have rephrased the sentence in the revised manuscript as follows:

No meta-analysis was performed for other secondary outcomes as only a single observational study reported on some of these outcomes. (Page 12, lines 252-253).

The study is well conducted and the conclusions drawn by the authors are plausible.

Response: We thank the reviewer for the comment.

Reviewer 2 Report

The article has a very good statistical analysis, but you should add more information.

For example, you should explain how often do you use Cannulas with long and narrow tubing (CLNT), what are the results you obtained by using them.

I am a neonatologist and I must confess that I haven't encountered the use of CLNT in any of the NICU  I visited. Also there are not many articles describing this method of ventilation.

You're article should contain more information about CLNT and should better describe the importance of using it.

Author Response

Response:

We thank the reviewer for the comment. We have incorporated some additional information regarding CLNT in the discussion section:

“The RAM cannula is quite similar to the traditional nasal prongs used to provide supplemental oxygen. Its long and narrow tubing is supposed to be more effective than traditional cannula in transmitting pressures to the nasal end which in turn is relatively softer compared to SBPs.”

Page 12, lines 272-275

“The softer nasal cannula which comes in contact with the nares is perceived to be associated with lesser nasal trauma compared to SBPs or NM”

Page 12, lines 282-285